# Meson formation in mixed-dimensional $t-J$ models

Fabian Grusdt[1,], Zheng Zhu[2], Tao Shi[3] and Eugene Demler[1]

**1** Department of Physics, Harvard University, Cambridge, Massachusetts 02138, USA
**2** Department of Physics, Massachusetts Institute of Technology,
Cambridge, Massachusetts 02139, USA
**3** CAS Key Laboratory of Theoretical Physics, Institute of Theoretical Physics, Chinese
Academy of Sciences, P.O. Box 2735, Beijing 100190, China

## Abstract

Surprising properties of doped Mott insulators are at the heart of many quantum materials, including transition metal oxides and organic materials. The key to unraveling complex phenomena observed in these systems lies in understanding the interplay of spin and charge degrees of freedom. One of the most debated questions concerns the nature of charge carriers in a background of fluctuating spins. To shed new light on this problem, we suggest a simplified model with mixed dimensionality, where holes move through a Mott insulator unidirectionally while spin exchange interactions are two dimensional. By studying individual holes in this system, we find direct evidence for the formation of mesonic bound states of holons and spinons, connected by a string of displaced spins – a precursor of the spin-charge separation obtained in the 1D limit of the model. Our predictions can be tested using ultracold atoms in a quantum gas microscope, allowing to directly image spinons and holons, and reveal the short-range hidden string order which we predict in this model.

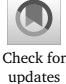

# 1 Introduction

The Fermi-Hubbard model represents one of the most fundamental and paradigmatic models of strongly correlated matter. It features an an intricate interplay of spin and charge degrees of freedom, expected to be relevant to high-temperature superconductivity observed in cuprate compounds [1–5]. However many basic features of of the Hubbard model remain poorly understood, which makes it challenging to identify the origin of such ubiquitous experimental phenomena as the non-Fermi liquid behavior [6], charge modulation [7], or the pseudogap [3, 8].

To approach this problem, here we propose to study a simplified model system which can be experimentally realized with, e.g., ultracold atoms. Instead of the two-dimensional (2D) $t-J$ model, which is commonly used to capture the interplay of spin and charge degrees of freedom in the low energy sector of the Hubbard model [3], we suggest to realize a system with mixed dimensionality: While the spin system is fully 2D, the holes doped into the system can only move along one direction, see Fig. 1 (a). On the one hand, this model shares many features with the 2D $t-J$ model, in particular the emergence of true long-range order in the ground state at zero doping. On the other hand, tuning the spatial anisotropy of the Heisenberg couplings allows us to study the transition to decoupled 1D chains, where spin and charge degrees of freedom separate. Moreover, being mappable to a problem of hard-core bosons, the model is sign-problem free, thus enabling efficient quantum Monte Carlo simulations for arbitrary doping values.

In this article we approach the mixed dimensional (mixD) $t-J$ model from the low-doping side and study the interplay of spin and charge degrees of freedom on the most fundamental level. To this end we consider individual holes doped into an antiferromagnet (AFM). In 2D, the single hole propagating through an AFM is commonly described by a magnetic polaron – a quasiparticle with a strongly renormalized dispersion due to the dressing with magnetic excitations [9–17]. While this description provides a powerful theoretical toolbox, it provides limited physical insight to the microscopic interplay of spin and charge excitations. More intuitive physical understanding can be gained by the parton construction put forward by Béran et al. [18]. These authors suggested that the single hole can be understood as a bound state of two partons: a neutral spinon and a spin-less holon. This closely resembles mesons formed by quark-antiquark pairs in high-energy physics. Recently it has been shown for the simplified $t-J_z$ model with reduced quantum fluctuations [19] that this phenomenology is

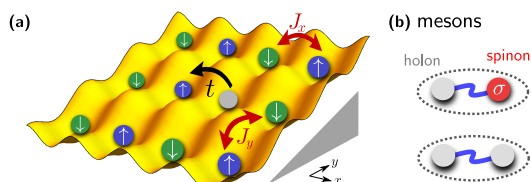

Figure 1: **Mixed-dimensional $t-J$ model.** We consider ultracold spin-1/2 fermions in an optical lattice at strong couplings. (a) By introducing a strong potential gradient along $y$-direction, the tunneling of holes with rate $t$ can be restricted to the $x$-axis, whereas $SU(2)$ invariant super-exchange interactions with tunable strengths $J_x$ and $J_y$ persist in both directions. We study the resulting mixD $t-J$ model in the low-doping regime and demonstrate that holes form mesonic bound states of spinons and holons (b), which can be directly observed using quantum gas microscopes. Mesons formed by pairs of holons have a higher energy, indicating the absence of strong pairing in mixD.

closely related to the string picture of magnetic polarons [20–25] and it can be justified on a microscopic level, enabling accurate quantitative predictions [26].

In 2D, direct observations of the strings and partons constituting magnetic polarons are challenging due to strong quantum fluctuations. Here, instead, we study holes in the mixD $t-J$ model. In this case we show that spinons and holons are connected by straight strings of displaced spins, making it easier to observe and characterize them. Even in the presence of quantum fluctuations of the surrounding spins, we demonstrate that the individual partons can be directly detected using experimatal tools available in systems of ultracold atoms in quantum gas microscopes [27–31].

By tuning the ratio of spin-exchange interactions along different lattice directions, our results in mixD can be related to the physics of 1D $t-J$ models. In a genuine 1D system, hole excitations decay into pairs of deconfined spinons and holons [32–34]. This fractionalization of the hole introduces quasi-long range non-local string order in the 1D system [35], which has recently been observed using a Fermi gas microscope [36]. These same measurements can be performed in the mixD $t-J$ model. In this case, we show that spinons and holons are confined and form bound states, see Fig. 1 (b). Hence, non-local string order emerges on a tunable length scale and should be readily observable experimentally. We also discuss the possibilities of stripe formation and pairing of holes in the mixD model.

## 2 Results

**Model.** We consider the mixD $t-J$ model of $S = 1/2$ fermions $\hat{c}_{i,\sigma}$, on lattice sites $i$ with spin $\sigma =\uparrow,\downarrow$, defined by the following Hamiltonian ($\hbar = 1$),

$$
\hat{\mathcal{H}} = \sum_{\langle i,j \rangle_x} \left[ -t \sum_\sigma \hat{\mathcal{P}}_{\text{GW}} \big( \hat{c}_{i,\sigma}^\dagger \hat{c}_{j,\sigma} + \text{h.c.} \big) \hat{\mathcal{P}}_{\text{GW}} + J_x \left( \hat{S}_i \cdot \hat{S}_j - \frac{\hat{n}_i \hat{n}_j}{4} \right) \right]
$$
$$
+ J_y \sum_{\langle i,j \rangle_y} \left( \hat{S}_i \cdot \hat{S}_j - \frac{\hat{n}_i \hat{n}_j}{4} \right). \tag{1}
$$

Here $\langle i,j \rangle_{x,y}$ denotes a pair of nearest neighbor (NN) sites in $x$ and $y$ directions, respectively, in a 2D square lattice, and every bond is counted once. The operator $\hat{\mathcal{P}}_{\text{GW}}$ denotes a Gutzwiller projection onto states with zero or one fermion per lattice site, and $\hat{n}_j$ and $\hat{S}_j$ are the fermion number and spin operators on site $j$.

Up to a next-nearest neighbor hole hopping term correlated with the surrounding spins [37], which is not expected to change physical properties significantly, Eq. (1) provides an accurate representation of the 2D Fermi-Hubbard model with a strong potential gradient $V(j) = -j_y \Delta$ in $y$-direction at strong couplings. In this way our model can be implemented using ultracold fermions in optical lattices [29–31]. When $U$ is the on-site interaction energy and $t$ the tunnel coupling between neighboring lattice sites, the super-exchange energies in $x$ and $y$ directions are

$$
J_x = \frac{4t^2}{U}, \qquad J_y = \frac{2t^2}{U+\Delta} + \frac{2t^2}{U-\Delta}, \tag{2}
$$

assuming that $|U|, |U \pm \Delta| \gg t$.

**Geometric strings, squeezed space and mesons.** In the following we restrict our discussion to a single hole, localized on the central chain where $j_y = 0$, in a system with net magnetization $S^z = 1/2$. We focus on the strong coupling limit $t \gg J_{x,y}$, where we argue that mesons form on intermediate length scales.

Our starting point is the ground state $|\Psi_0\rangle$ of the 2D Heisenberg model without a hole and with total spin $S^z = 0$. To construct a set of relevant basis states including the hole, we remove a

spin-down particle at site $(j_x^s, 0)$ and obtain the state $\hat{c}_{j_x^s,0,\downarrow}|\Psi_0\rangle \equiv |\psi_0\rangle|j_x^s\rangle$, where $|\psi_0\rangle$ denotes a pure state of spins on the lattice sites $\tilde{j} \neq (j_x^s, 0)$. Because the hopping $t$ is the largest energy scale, we start by constructing all allowed states that can be reached by applying the hopping part $\hat{\mathcal{H}}_t$ of the Hamiltonian, defined by terms proportional to $t$ in Eq. (1). Because the hole can only move on the central chain, these states can be labeled by the distance $\Sigma = j_x - j_x^s$ of the hole at site $j_x$ from the original site $j_x^s$, and we denote these orthonormal states by $|\psi_0\rangle|j_x^s, \Sigma\rangle$.

The difficulty of the $t-J$ model stems from the fact that the hole motion distorts the surrounding spin state. In the approximate set of basis states constructed so far this corresponds to a displacement of all spins along $\Sigma$, referred to as the geometric string, connecting $j_x$ and $j_x^s$. More generally, we can label the spins by their original positions $\tilde{j}$ in the lattice before the hole was created. In analogy with 1D, see Refs. [35, 38], we call the space defined by the spins on these lattice sites $\tilde{j} \neq (j_x^s, 0)$ squeezed space. The key advantage of the new labeling is that the hole motion has no effect on the configuration of spins in squeezed space. Instead, the geometry of the couplings between spins in squeezed space is modified along the geometric string $\Sigma$ (see Methods for details).

To formulate the Hamiltonian (1) projected in the truncated basis, we introduce bosonic holon operators for which $\hat{h}_\Sigma^\dagger|0\rangle = |\Sigma\rangle$. The hopping part of the Hamiltonian becomes $\hat{\mathcal{H}}_t = -t \sum_{\langle\Sigma',\Sigma\rangle}(\hat{h}_{\Sigma'}^\dagger \hat{h}_\Sigma + \text{h.c.})$. When $t \gg J_{x,y}$, quantum correlations between the strongly fluctuating string $\Sigma$ and spins in squeezed space can be neglected. In the simplest, so-called frozen spin approximation (FSA) we can assume that the spin wavefunction $|\psi_0\rangle$ in squeezed space does not change upon doping and the single-hole wavefunction takes a product form $|\Psi\rangle \approx |\psi_0\rangle|\phi_\Sigma\rangle$, where $|\phi_\Sigma\rangle$ describes the holon. We will confirm below, in Fig. 4, that the FSA is a reliable approximation.

Within the FSA, terms in Eq. (1) proportional to $J_{x,y}$ give rise to an effective potential [26, 39] depending on the length of the geometric string $\ell_\Sigma = |\Sigma|$,

$$\hat{\mathcal{H}}_J = \sum_\Sigma \hat{h}_\Sigma^\dagger \hat{h}_\Sigma \left(\frac{dE}{d\ell}\ell_\Sigma + g_0\delta_{\ell_\Sigma,0} + \mu_{\text{h}}\right). \tag{3}$$

It depends only on spin correlators in the wavefunction $|\Psi_0\rangle$ without the hole: $dE/d\ell = 2J_y(C_2 - C_1^y)$, $g_0 = -J_x(C_3^x - C_1^x)$ and $\mu_{\text{h}} = J_x(1/2 + C_3^x - 3C_1^x) + J_y(1/2 - 2C_1^y)$, where $C_1^\mu = \langle\Psi_0|\hat{\boldsymbol{S}}_j \cdot \hat{\boldsymbol{S}}_{j+e_\mu}|\Psi_0\rangle$ for $\mu = x, y$, $C_2 = \langle\Psi_0|\hat{\boldsymbol{S}}_j \cdot \hat{\boldsymbol{S}}_{j+e_x+e_y}|\Psi_0\rangle$ and $C_3^x = \langle\Psi_0|\hat{\boldsymbol{S}}_j \cdot \hat{\boldsymbol{S}}_{j+2e_x}|\Psi_0\rangle$.

Because Eq. (3) contains a linear confining potential $\propto \ell_\Sigma$, the holon is bound to the lattice site $\boldsymbol{j}^s = (j_x^s, 0)$ where it was initially created. Due to spin exchanges this lattice site $\boldsymbol{j}^s$ will develop dynamics on its own, but on a time scale $1/J$ larger than $1/t$ on which the holon motion takes place. The string tension $dE/d\ell$ depends only on the local correlators $C_1^x$, $C_2$ but does not require long-range order. The average string length in the bound state scales as $(t/J_y)^{1/3}$ when $t \gg dE/d\ell$ [20].

Physically, this bound state can be understood as a meson formed by a spin-less holon and a charge-neutral spinon, which are connected by the geometric string $\Sigma$ of displaced spins [18, 26]. In general, the end of the string at lattice site $\boldsymbol{j}^s$ corresponds to a geometric defect in real space. Since it was initially created from $|\Psi_0\rangle$ by removing a spin-down fermion, it can be associated with a spin $S^z = 1/2$, thus corresponding to a spinon excitation. Because there exist no deconfined spinons in the 2D Heisenberg AFM, the geometric defect and the spinon are expected to form a stable bound state.

Based on our theoretical analysis so far, we can construct a variational wavefunction of mesons in the mixD $t-J$ model which also includes spinon dynamics. To this end we start from a representation of the 2D Heisenberg AFM by slave fermions $\hat{f}_{\boldsymbol{k},\sigma}$ [40] and approximate the ground state wavefunction as $|\Psi_0\rangle = \hat{\mathcal{P}}_{\text{GW}}|\Psi_{\text{MF}}\rangle$. Here the MF wavefunction $|\Psi_{\text{MF}}\rangle = \prod_{\boldsymbol{k}\in\text{MBZ}} \hat{f}_{\boldsymbol{k},\uparrow}^\dagger \hat{f}_{\boldsymbol{k},\downarrow}^\dagger|0\rangle$ describes a band insulator obtained by spin-1/2 fermions hopping

on a square lattice with staggered flux $\Phi = \pm 0.4\pi$ per plaquette and a staggered magnetic field, of strength $B_{st} = 0.44$ in units of the hopping, breaking the $SU(2)$ symmetry [41, 42].

A spinon-holon pair excitation with spin $\sigma$ can be created at site $j^s$ by the operator $\hat{c}_{j^s,\overline{\sigma}} = \hat{h}_{j^s}^\dagger \hat{f}_{j^s,\overline{\sigma}}$ where $\overline{\uparrow} = \downarrow$ and $\overline{\downarrow} = \uparrow$. To take into account the holon motion, which creates the geometric string, we propose the following trial wavefunction for the meson:

$$|\Psi_{MP}\rangle = \mathcal{N} \sum_{j^s} e^{-i\boldsymbol{k}_{MP} \cdot \boldsymbol{j}^s} \sum_\Sigma \phi_\Sigma \hat{G}_\Sigma \hat{\mathcal{P}}_{GW} \hat{f}_{j^s,\overline{\sigma}} |\Psi_{MF}\rangle. \tag{4}$$

Here $\phi_\Sigma$ is the string wavefunction, which in practice we determine from the effective model in Eq. (3). The operator $\hat{G}_\Sigma$ acts on Fock states with one empty site – the holon position – from where it starts to create the geometric string $\Sigma$ by displacing fermions along $\Sigma$. Because the meson wavefunction (4) includes the string, which binds spinons to holons, it is markedly different from resonating valence bond states commonly used for approximate descriptions of the $t - J$ model at finite doping. Finally, $\boldsymbol{k}_{MP}$ denotes the center of mass momentum of the meson, which is carried by the heavy spinon.

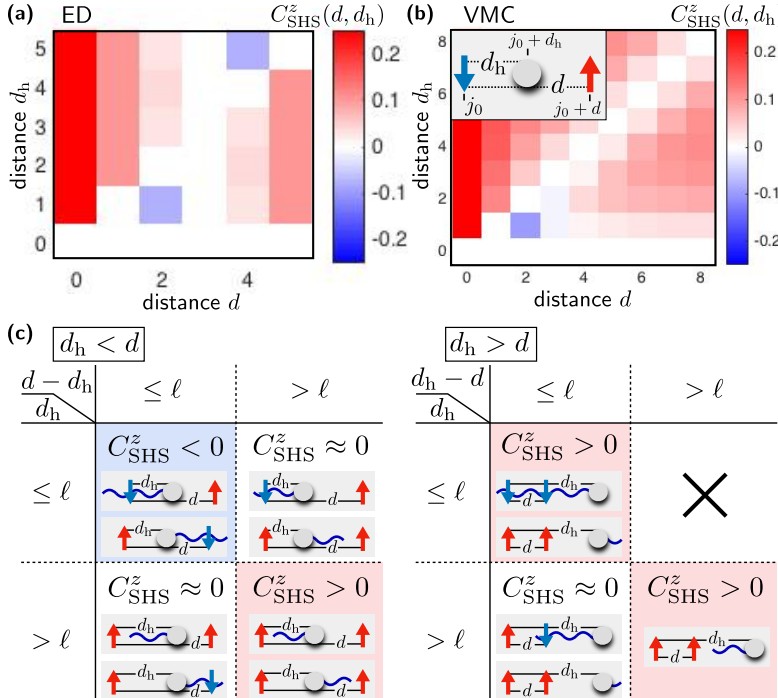

Figure 2: **Signatures of meson formation**. We calculate spin-hole-spin correlations for $t = 3J$. (a) The three-point function $C_{SHS}^z(d, d_h)$, calculated using ED in a $6 \times 3$ system periodic along $x$, changes sign at $d = 3$. This indicates the presence of a geometric string with an average length around one. (b) The same behavior is predicted in a $16 \times 8$ periodic system by the trial wavefunction in Eq. (4) which we evaluated at $\boldsymbol{k}_{MP} = (\pi/2, \pi/2)$, $\Phi = \pm 0.4\pi$, $B_{st} = 0.44$ using VMC methods. (c) The structure of $C_{SHS}^z(d, d_h)$ observed in (a) and (b) can be understood from the string picture by comparing the distances $d_h$, $d - d_h$ and $d_h - d$ to the typical string length $\ell$ and distinguishing the two cases $d_h < d$ and $d_h > d$. Purple wavy lines are schematic representations of the strings. Spin correlations change sign when crossing either end of the string. Blue spins are part of the geometric string and have changed the sublattice, introducing negative signs in $C_{SHS}^z(d, d_h)$. One has to average over the opposite orientations of the string relative to the hole, $\Sigma = \pm \ell$, to estimate the value of the correlator.

**Signatures for meson formation.** The trial wavefunctions we discussed so far factorize in squeezed space. Nevertheless they describe strongly correlated states in the mixD $t-J$ model, since physical observables in real space depend explicitly on the instantaneous string configuration. Now we present numerical simulations which support the meson picture and confirm the accuracy of the trial wavefunction (4).

Signatures of meson formation can be obtained directly from spin-charge correlations. We start by considering the three-point function which has been measured in the 1D Fermi-Hubbard model in Ref. [36],

$$C_{\text{SHS}}^z(d, d_{\text{h}}) = (-1)^d \langle \hat{S}_{j_0}^z \hat{n}_{j_0+d_{\text{h}}}^{\text{h}} \hat{S}_{j_0+d}^z \rangle / \langle \hat{n}_{j_0+d_{\text{h}}}^{\text{h}} \rangle. \tag{5}$$

Here $\hat{n}_j^{\text{h}}$ denotes the hole density, we assume that the $y$-indices of all operators are $j_y = 0$, and $j_0$ is an arbitrary reference site, see Fig. 2.

When the distance of the spin at $j_0$ to the hole is smaller than the distance to the second spin, $d_{\text{h}} < d$, the spin-correlator is taken across the hole. If, in addition, $d - d_{\text{h}} \leq \ell$ and $d_{\text{h}} \leq \ell$ where $\ell$ is the string length, one of the two spins is always part of the geometric string, while the other is not. Since the spins on the string have switched sublattice, we expect that the correlator $C_{\text{SHS}}^z < 0$ has a non-trivial sign in this case. Otherwise the correlations are suppressed, $C_{\text{SHS}}^z \approx 0$, or $C_{\text{SHS}}^z > 0$ shows AFM correlations, see Fig. 2 (c).

These expectations obtained from the FSA are confirmed by numerical ED simulations in Fig. 2 (a). In particular we find that $C_{\text{SHS}}^z$ changes sign at $d = 3$ for $d_{\text{h}} = 1$, consistent with the expected average string length $\langle \phi_\Sigma | \hat{\ell} | \phi_\Sigma \rangle = 0.74$ at the considered value of $t/J = 3$, where $J = J_x = J_y$. The ED results are also in excellent agreement with predictions by the trial wavefunction from Eq. (4), which we evaluated using variational Monte Carlo (VMC) techniques [43], see Fig. 2 (b). Our results can be tested using ultracold fermions by repeating previous measurements performed in 1D [36] in the mixD setting. Finite temperatures lead to a decreased magnitude $|C_{\text{SHS}}^z(d, d_{\text{h}})|$, but we expect that the sign change of $C_{\text{SHS}}^z$ is robust at moderate temperatures $T \lesssim J$.

In systems with long-range AFM order [44], i.e. $\langle \hat{S}_j^z \rangle = \Omega_j (-1)^{j_x+j_y}$ where $\Omega_j$ is the AFM order parameter, indications that holons bind to spinons can also be found in the two-point spin-hole correlator

$$C_{\text{SH}}^z(d_{\text{h}}) = (-1)^{d_{\text{h}}+j_0} \left[ \langle \hat{S}_{j_0+d_{\text{h}}}^z \hat{n}_{j_0}^{\text{h}} \rangle - \langle \hat{S}_{j_0+1+d_{\text{h}}}^z \hat{n}_{j_0+1}^{\text{h}} \rangle \right]. \tag{6}$$

As in Eq. (5) we assume that all spin operators are evaluated on the central chain, i.e. $j_y = 0$, and $j_0$ denotes a reference site. Note that $C_{\text{SH}}^z$ is defined as a sum of two terms, which contribute with opposite signs and correspond to holes on different sublattices. This cancels weak residual oscillations of the individual terms with $d_{\text{h}}$, originating from imbalanced hole populations in the two sublattices related to the spin quantum number of the spinon. The latter is fixed to $S^z = 1/2$ in our case because we restrict our numerical analysis to systems with net magnetization $S^z = 1/2$. Experimentally, the three-point function in Eq. (5) is advantageous, because in contrast to $C_{\text{SH}}^z(d_{\text{h}})$ it does not depend sensitively on the net magnetization, which varies from shot to shot [36].

In Fig. 3 (a) we show results for $C_{\text{SH}}^z(d_{\text{h}})$ calculated using DMRG in a three-leg ladder with a hole on the central leg. To mimic the effect of long-range AFM order expected in 2D, we added a staggered magnetic field $(-1)^{j_x+j_y} B \hat{S}_j^z$ on the outermost sites, see Fig. 3 (b), pinning the AFM order. We find a pronounced suppression of $C_{\text{SH}}^z(d_{\text{h}})$ at small $d_{\text{h}}$. This can be understood from the string picture by considering separately the cases (i) when $d_{\text{h}} > \ell$ exceeds the string length $\ell$, and (ii) when $d_{\text{h}} \leq \ell$, as illustrated in Fig. 3 (c).

In case (i), the spin at site $j_{\text{h}} + d_{\text{h}}$ is not part of the geometric string $\Sigma$ and we expect that $C_{\text{SH}}^z(d_{\text{h}}) \approx \Omega_0$ is related to the AFM order parameter $\Omega_0$ in the undoped system. In case

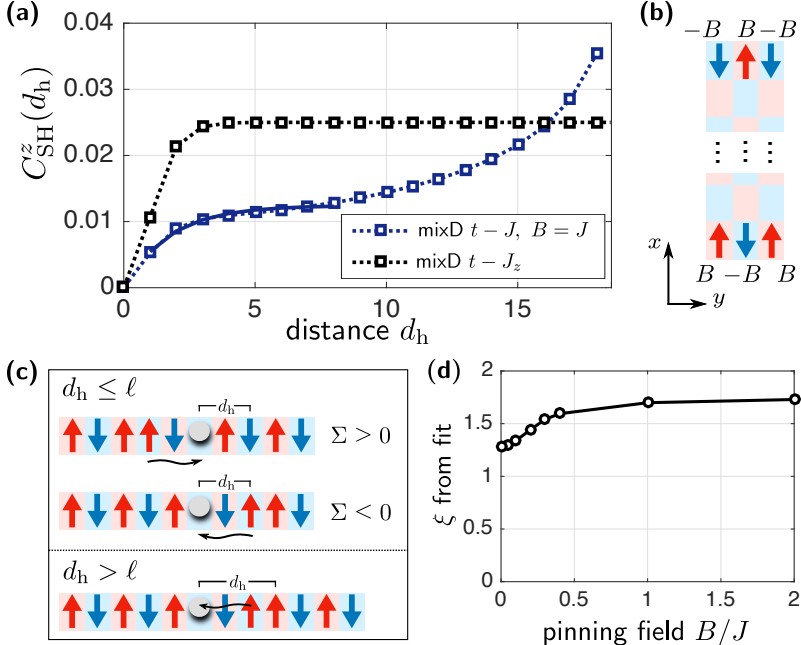

Figure 3: **Distortion of the Néel state by mesons**. We calculate spin-hole correlations for $t = 3J$. (a) The two-point function $C_{\text{SH}}^z(d_{\text{h}})$, calculated using DMRG in a $40 \times 3$ system with open boundary conditions for $j_0 = L_x/2$, shows a pronounced dip at small $d_{\text{h}}$. This is an indicator for meson formation, as can be understood by considering a hole moving in a classical Néel state (mixD $t-J_z$ model): (c) When the string length $\ell = |\Sigma| \geq d_{\text{h}}$, configurations with positive and negative strings, $\Sigma < 0$ and $\Sigma > 0$, cancel each other. When $\ell < d_{\text{h}}$ both configurations contribute with the same sign. The DMRG data in (a) includes a staggered magnetic field $B$ at the edges, as shown in (b), which pins the AFM order in the three-leg system. (d) The characteristic length $\xi$ of the dip in $C_{\text{SH}}^z(d_{\text{h}})$ at short distances changes only weakly when the staggered field is varied.

(ii), we distinguish between two additional configurations: when $\Sigma < 0$ ($\Sigma > 0$) the holon is located at the left (right) end of the string and the spin at site $j_{\text{h}} + d_{\text{h}}$ is (is not) part of the geometric string. By averaging the contributions $\pm\Omega_0$ from these two string orientations, which are equally likely due to inversion symmetry, we expect a reduction of $C_{\text{SH}}^z(d_{\text{h}})$ when $d_{\text{h}} \leq \ell$.

The width of the dip in $C_{\text{SH}}^z(d_{\text{h}})$ around $d_{\text{h}} = 0$ characterizes the typical string length, i.e. the size of the meson. To extract it, we fit $C_{\text{SH}}^z(d_{\text{h}})$ by a function $A_1 + A_2 e^{-d_{\text{h}}/\xi}$ in the regime $d_{\text{h}} = 1, ..., 8$ as indicated by a solid line in Fig. 3 (a). The fitted values $\xi(B)$ are shown in Fig. 3 (d) as a function of the pinning field $B$. As expected from the FSA, $\xi(B)$ shows only weak dependence on $B$ and is on the order of one lattice site. Notably, this remains true for $B = 0$, where the undoped three-leg ladder has no long-range order and belongs to the same universality class as a 1D spin-1/2 chain [45].

**Direct imaging of geometric strings.** The indicators of meson formation discussed so far are based on expectation values of two- and three-point operators. The single-site resolution achieved by quantum gas microscopes allows to determine these quantities by averaging over multiple measurements in the $z$-basis of the spins. Even more information can be extracted by analyzing the individual experimental snapshots. For example, it has been demonstrated that this allows to measure string order [36, 46], or the full counting statistics of the staggered

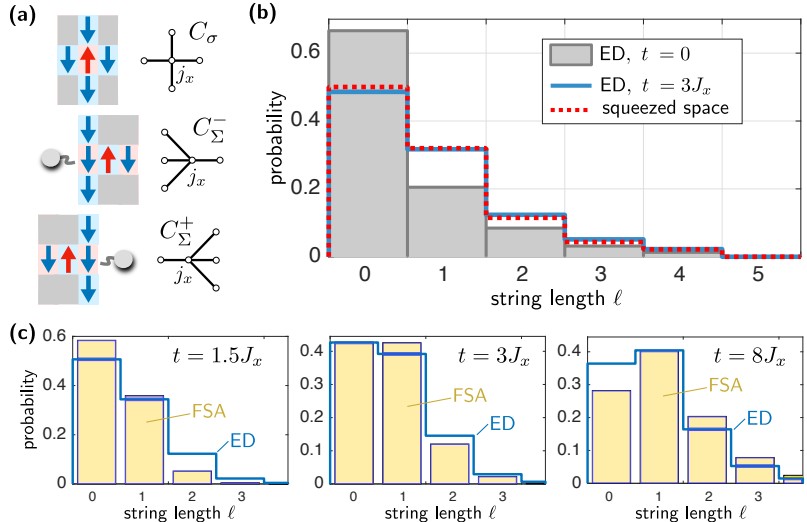

Figure 4: **Signatures of geometric strings.** We calculate the full counting statistics of the string length which can be measured in a quantum gas microscope. (a) The correlators $C_\sigma(j_x)$ and $C_\Sigma^\pm(j_x)$ defined in Eqs. (7), (8) can be used to extract the configuration of the geometric string at site $(j_x, 0)$ for individual measurements in the $z$-basis. (b) From the extracted string configurations, we determine the full counting statistics of the string length $\ell$. We used ED in a $6 \times 3$ system with $J_x = J_y = J$ at $t = 0$ and $t = 3J$. Our results are compared to predictions from the FSA in squeezed space (dotted line), as explained in the main text. (c) We compare the string length distribution derived from FSA ($p_{\text{FSA}}(\ell)$ defined in the main text, bar plots) to the distribution extracted from snapshots of the ground state wavefunction (determined from ED, line plots). The ED results are obtained as in (b), but here we post-selected states with more than 50% of the maximum staggered magnetization.

magnetization [44]. Now we show that hidden string order related to meson formation is also observable in the mixD $t - J$ model.

In order to determine the string configuration from a given snapshot, we consider the following operators,

$$\hat{C}_\sigma(j_x) = \sum_{\boldsymbol{i} = \text{NN of } (j_x, 0)} \hat{S}_{j_x, 0}^z \hat{S}_{\boldsymbol{i}}^z, \tag{7}$$

$$\hat{C}_\Sigma^\pm(j_x) = \sum_{\delta j = -1, +1} \hat{S}_{j_x, 0}^z (\hat{S}_{j_x + \delta j, 0}^z + \hat{S}_{j_x \pm 1, \delta j}^z). \tag{8}$$

As shown in Fig. 4 (a), $\hat{C}_\sigma(j_x)$ measures NN correlators $C_1$ in real space if $j_x$ is not part of the geometric string. Similarly, the correlator $\hat{C}_\Sigma^\pm(j_x)$ measures NN correlators $C_1$ in squeezed space if $j_x$ is part of a geometric string with the holon located at its right $(+)$ or left $(-)$ end, respectively. Therefore we expect that a geometric string is present at site $j_x$ if the measured value of $\hat{C}_\sigma(j_x)$ in the snapshot is larger than the measured values of $\hat{C}_\Sigma^\pm(j_x)$.

By comparing the values of the three correlators defined in Eqs. (7), (8) we can extract a likely configuration of the geometric string in every individual shot of the measurement (see Methods for more details). We emphasize that due to quantum fluctuations it is impossible to reconstruct the exact string configuration, but for our purposes it will be sufficient that we found a reasonable proxy for the latter.

From the extracted configuration we can easily determine the string length $\ell$ in every individual shot. In Fig. 4 (b) we plot the full counting statistics of the string lengths $\ell$. To this end, we generated snapshots by exact diagonalization (ED) of a $6 \times 3$ system with periodic

boundary conditions along $x$ and $S^z = 1/2$. When the hole is pinned, $t = 0$, we only find a few short strings caused by quantum fluctuations of the spins.

When we increase the tunneling to $t = 3J$, we observe a clear increase of the number of strings with lengths $\ell = 1, 2, 3$. To compare this result with our expectations from the FSA in squeezed space, we start from the snapshots at $t = 0$ and construct a new set of configurations $\{|\alpha'\rangle\}$ by including the hole motion by hand (i.e. we apply the operator $\hat{G}_\Sigma$ from Eq. (4)). Here the string length $\ell = |\Sigma|$ is chosen randomly from the distribution given by the string wavefunction $|\phi_\Sigma|^2$ which we calculate using the FSA. By extracting the string configurations from the new shots $\{|\alpha'\rangle\}$ as before, we obtain the string length distribution shown by a dotted line in Fig. 4 (b). This result, constructed from FSA, agrees remarkably well with the exact distribution function obtained directly from ED at $t = 3J$ and supports the meson theory of holes in the mixD $t - J$ model.

To test the accuracy of the FSA further, we calculate the distribution function of the extracted string lengths for different $t/J$ in Fig. 4 (c). To reduce the effects of quantum fluctuations, we considered only shots with a total staggered magnetization above 50% of its maximum value. We have checked that the squeezed space construction, starting from snapshots at $t = 0$, still yields excellent agreement in this case. In Fig. 4 (c) we provide a direct comparison of the obtained string length distribution with the FSA result $p_{\text{FSA}}(\ell) = \delta_{\ell,0}|\phi_{\Sigma=0}|^2 + 2(1 - \delta_{\ell,0})|\phi_{\Sigma=\ell}|^2$ (bar plot in Fig. 4 (c)). Although complete quantitative agreement is still not expected due to residual quantum fluctuations, we observe that the string lengths extracted from our ED simulations show the same qualitative features as predicted by the FSA: For small values of $t/J$ we obtain a pronounced maximum at $\ell = 0$, which becomes a plateau at $\ell = 0, 1$ for $t/J = 3$ and develops into a dip at $\ell = 0$ when $t \gg J$.

**Dimensional crossover.** Our numerical analysis so far was restricted to spatially isotropic couplings, $J_x = J_y$. Now we study the dimensional crossover by tuning $J_y/J_x$. In the 1D limit, $J_y = 0$, the string tension $dE/d\ell = 0$ vanishes and it is well-known that spinons and holons are deconfined [32]. This leads to geometric strings extending over the entire length of the system [35, 38], which have been observed experimentally in Ref. [36]. Because the string tension is finite when $J_y > 0$, we expect that the average string length $\langle \hat{\ell} \rangle$ diverges at $J_y = 0$.

In Fig. 5 (a) we plot the string length $\xi$ extracted from fits of the spin-hole correlators for various $J_y/J_x$. We used DMRG to obtain the two-point function $C_{\text{SH}}^z(d_{\text{h}})$ in a $40 \times 3$ system as in Fig. 2 and calculated the three-point function $C_{\text{SHS}}^z(d, d_{\text{h}})$ at $d = 1$ from the trial wavefunction in Eq. (4) using VMC methods. Both approaches show an increase of the string length when $J_y$ approaches zero, and the DMRG data points in the range $0.2 \leq J_y \leq 1$ are well described by a power law $\xi(J_y) \approx 1.3 \times (J_y/J_x)^{-0.9}$.

From the meson wavefunction (4) we obtain shorter string lengths than predicted by DMRG. We expect that this is due to an inaccuracy of the FSA string potential Eq. (3), which contains a weak local spinon-holon attraction $g_0$. The latter results from the oversimplified description of the spinon in FSA as a missing spin, which is inaccurate in 1D, and leads to a spinon-holon bound state with a large but finite binding length for $J_y = 0$.

To check the accuracy of the trial wavefunction (4), we calculate its variational energy $\langle \Psi_{\text{MP}} | \hat{\mathcal{H}} | \Psi_{\text{MP}} \rangle$ in Fig. 5 (b) and compare it to our DMRG results. Qualitatively we obtain similar behavior as a function of $J_y$, although the variational energy is larger than the DMRG result by an amount of order $J_y$. We expect that the dominant factors contributing to this discrepancy are (i) the use of only straight strings along $x$ in $|\Psi_{\text{MP}}\rangle$ and (ii) our neglect of spin-hole correlations in squeezed space. Both effects should lead to corrections of order $J_y$. More details of our analysis of the crossover are provided in the Methods.

**Precursors of stripe formation.** In Fig. 6 (a), (b) we use DMRG simulations to study spin and

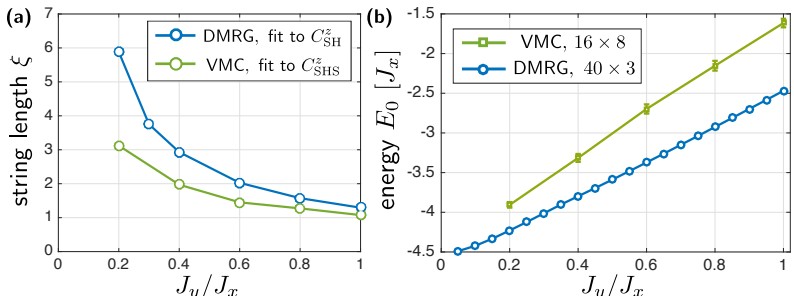

Figure 5: **Dimensional crossover.** We change from a 1D to a mixD situation by tuning $J_y/J_x$ at $t = 3J_x$. (a) The string length $\xi$, extracted by fits of spin-hole correlators $C_{\mathrm{SH}}^z$, sharply increases when $J_y \to 0$. This indicates a deconfinement of the spinon-holon pair in this limit. (b) We compare the ground state energy $E_0$ of a single hole from DMRG in a $40 \times 3$ system to the variational energy of the trial wavefunction in Eq. (4) in a $16 \times 8$ system, evaluated at $\boldsymbol{k}_{\mathrm{MP}} = (\pi/2, \pi/2)$, $\Phi = 0.5\pi$ using VMC methods, see Methods for more details.

charge orders in finite-size mixD systems with open boundaries and total spin $S^z = 1/2$. We observe a pronounced maximum of the hole density $n_{j_x}^{\mathrm{h}}$ in the center, which is accompanied by a sign change of the surrounding Néel order $\Omega_j = (-1)^{j_x+j_y} \langle \hat{S}_j^z \rangle$. Such behavior also occurs in the stripe phase of cuprates [47], where the Néel order changes sign across a line of enhanced hole density.

Here we interpret these features as precursors of stripe formation in finite $n$-leg ladders. In larger systems with the same hole doping in every $n$-th chain we expect, similarly, to observe robust stripes. We note that the stripe features are absent in our simulations when periodic boundary conditions and even numbers of lattice sites are used along $x$, e.g. in Fig. 2 (a). In the limit of a single hole in an infinite system we also expect these features to disappear, because it is energetically unfavorable to sustain a 1D line defect where $\Omega_j$ changes sign.

Explaining the formation of stripe-like structures in finite-size systems requires a modification of the meson theory. In Fig. 6 (c) we provide some intuition by considering spins in a classical Néel background. We note that the Néel order can only change sign across the hole, if a domain wall of two aligned spins is present in chains without a hole. Therefore the stripe-like ground state we found in the three-leg ladder can be understood as a baryonic bound state of two spinons and one holon, see Fig. 6 (d). We expect that the motion of the holon still leads to the formation of geometric strings, connecting it to the two spinons.

Similarly, we expect that the ground state of the finite-size five-leg system, see Fig. 6 (b), corresponds to a "petaquark" state formed by one holon bound to four spinons, see Fig. 6 (d). A detailed investigation of the dimensional cross-over from 3-to-5-to...-$2n + 1$ leg setting to the infinite mixD system will be subject of future work.

## 3   Discussion

In this article we propose a simplified model to study some of the exotic phenomena expected to play a fundamental role in the ground state of the 2D Fermi-Hubbard model, and high-$T_c$ cuprate superconductors. As a key simplification we consider holes which can only move along one direction, described by the mixD $t-J$ model. Our model Hamiltonian can be implemented at arbitrary doping levels using ultracold atoms in optical lattices.

We study this model at low doping, and provide evidence that holes form mesonic bound

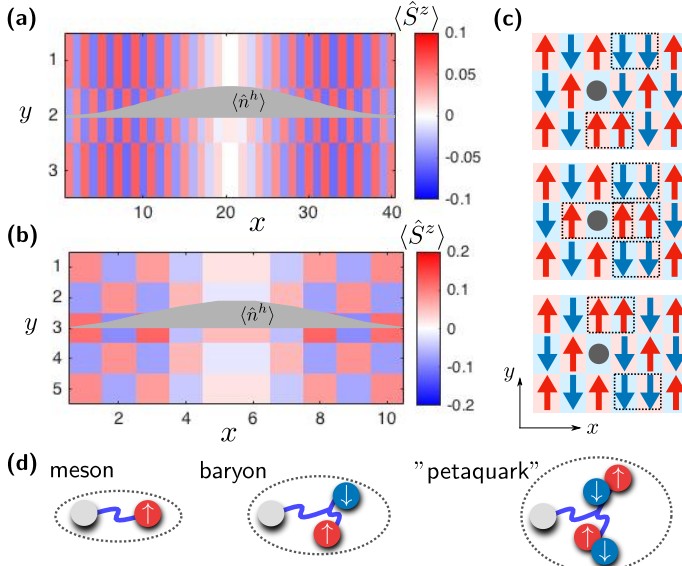

Figure 6: **Precoursors of stripe formation**. We consider a single hole moving on the central chain of an $n$-leg ladder, described by the mixD $t-J$ model with with open boundary conditions. We performed DMRG simulations for (a) an $n = 3$-leg system with $L_x = 40$ sites in $x$-direction and (b) $n = 5$ and $L_x = 10$. The hole density $\langle \hat{n}^{\mathrm{h}}_j \rangle$ forms a standing wave pattern with a pronounced maximum in the center of the middle chain (gray). The AFM order parameter $\Omega_j = (-1)^{j_x + j_y} \langle \hat{S}^z_j \rangle$ changes sign in the center around $j_x = L_x/2$. To understand this pattern from a parton perspective, one needs to consider multi-parton bound states. (c) In a three-leg ladder, the holon is bound to two spinons (domain walls) in the upper and lower chains. The possible spinon configurations (top and bottom) are coupled via a third state (center) with a spin-1 magnon excitation in the central chain. (d) The multi-parton state in the three-leg ladder is a generalization of a meson and can be understood as the analogue of a baryon. Similarly, the bound state in a five-leg ladder corresponds to a petaquark state.

states of spin-less holons and charge-neutral spinons. To model the structure of the mesons, we introduce a restricted set of basis states describing geometric strings of displaced spins which connect spinons and holons. We show that non-local spin-charge correlations provide evidence for meson formation, and demonstrate that geometric strings can be directly imaged in individual experimental snapshots. Our predictions can be tested in current experiments with ultracold atoms [29–31, 48, 49].

To check if two holes from the same leg can pair at zero temperature, we calculate the binding energy $E_{\mathrm{bdg}} = E_{2\mathrm{h}} - 2E_{1\mathrm{h}} + E_{0\mathrm{h}}$ using DMRG. Finite-size scaling for a mixD three-leg ladder with two holes in the central chain at $t = 3J$, up to lengths $L_x = 80$, extrapolates to $|E_{\mathrm{bdg}}| \approx 10^{-3}J$ when $1/L_x \to 0$, indicating the absence of strong pairing. In a forthcoming work we study meta-stable holon-holon mesons at higher energies, as illustrated in Fig. 1 (b). Because $|E_{\mathrm{bdg}}| \ll J$ we expect that they can decay into spinon-holon mesons by spontaneously creating spinon-antispinon pairs as in the Schwinger mechanism. Such dynamics can be studied experimentally using quantum gas microscopes.

While our calculations indicate that pairing is suppressed in mixD, we find precursors for the formation of stripe phases already for a single hole in a finite-size system. Simulations at higher doping values will be devoted to future work, but we expect that the mixD $t-J$ model can provide new insights into the interplay of superconductivity and stripe phases. Note that

the mixD Hamiltonian has many independent sectors of individually tunable doping levels per chain, which need to be studied separately. At finite doping we also expect that the relation of our meson approach with the fractionalized Fermi liquid theory of the pseudogap phase [50] or the phase string effect [51–53] can be explored. Finally, the goal is to extend our work and search for string patterns also in the 2D $t-J$ model [44, 54, 55]. An interesting starting point for the study of the mixD-to-2D crossover is the fate of the Nagaoka effect: While the ground state of the 2D $t-J$ model at $J = 0$ has ferromagnetic order [56, 57], it is highly degenerate in mixD.

# 4 Methods

**Mesons and squeezed space in mixD.** In the main text we describe spinon-holon mesons by the truncated string basis. It is obtained by first creating a hole at site $(j_x^s, 0)$ in the ground state $|\Psi_0\rangle$ of the undoped Heisenberg model, leading to the state $\hat{c}_{j_x^s, 0, \downarrow} |\Psi_0\rangle \equiv |\psi_0\rangle |j_x^s\rangle$. Next one applies the hopping part $\hat{\mathcal{H}}_t$ of the Hamiltonian (1) multiple times to generate a set of geometric string basis states, $|\psi_0\rangle |j_x^s, \Sigma\rangle$. These states describe a meson with a spinon localized at site $(j_x^s, 0)$. The displacement of the surrounding spins along the geometric string is taken into account, but otherwise their configuration is fixed by $|\psi_0\rangle$, determined from the undoped ground state.

Now we explain how these limitations of the truncated basis can be overcome and how changes in the spin wavefunction affect the physical picture. We distinguish between two types of processes: (i) The first type involves the lattice site $j^s$ associated with the spinon; It introduces spinon dynamics. (ii) The second type involves other fluctuations in the spin background; It leads to additional polaronic dressing of the meson.

To describe (i) we start by noting that the restricted string basis can be constructed for arbitrary initial positions of the hole, $j^s$ and $j^s + \delta j_x e_x$. The resulting basis states, which correspond to different spinon positions, are no longer orthogonal in general. However, it can be expected that they are approximately orthogonal as long as the undoped ground state $|\Psi_0\rangle$ has strong AFM correlations. In the case of the classical Néel state, this assumption becomes exact. Otherwise, the basis can be orthonormalized using the Gram-Schmidt method.

In general, we expect that the Hamiltonian has non-zero matrix elements between states corresponding to different spinon positions, $\langle j_x^s + \delta j_x, 0 | \langle \psi_0 | \hat{\mathcal{H}}_J | \psi_0 \rangle | j_x^s, 0 \rangle \neq 0$; Note that these states also correspond to different string configurations, but they must have the same holon position to guarantee a non-zero matrix element. The additional terms added to the effective Hamiltonian introduce spinon – and thus meson – dynamics. Because the matrix elements responsible for such processes are proportional to $J_x$, we expect that the typical spinon or meson bandwidth is proportional to $J_x \ll t$. The same result is predicted by conventional theories of magnetic polarons in 2D [9, 11, 12]; But in that case the severe modification of the bandwidth of the hole, from $8t$ for a free holon to an expression $\propto J_x$, is usually interpreted as a consequence of strong polaronic mass renormalization. In the main part of the paper, we have implicitly included spinon dynamics in the trial wavefunction in Eq. (4).

The second types of processes (ii) lead to polaronic dressing of the meson by spin-wave excitations. Now we argue that this can be understood as a result of quantum fluctuations of the surrounding spins. To describe such fluctuations, we introduce a generalization of the squeezed space commonly used to describe the 1D $t-J$ model [35, 38]. In 2D, the squeezed space can be constructed as an extension of the restricted string basis, assuming a fixed spinon position $j^s$. A new set of operators $\tilde{S}_{\tilde{j}}$ is defined on the squeezed space lattice, which is obtained from the original 2D lattice by excluding the site $(j_x^s, 0)$ where the hole was initially created.

In squeezed space, the hole motion has no effect, because the new operators $\tilde{S}_{\tilde{j}}$ explicitly

depend on the string configuration $\Sigma$. They can be defined by writing the operators $\hat{\boldsymbol{S}}_j$ on the original 2D lattice as

$$\hat{\boldsymbol{S}}_j = \sum_{\Sigma=-\infty}^{\infty} |\Sigma\rangle\langle\Sigma| \, \tilde{\boldsymbol{S}}_{g_\Sigma(j)}. \tag{9}$$

The sites $\boldsymbol{j}$ and $\tilde{\boldsymbol{j}} = g_\Sigma(\boldsymbol{j})$ in real and squeezed space are related by a string-dependent function $g_\Sigma(\boldsymbol{j})$ taking the role of a metric. This metric is defined by

$$\boldsymbol{g}_\Sigma(j_x, 0) = (j_x + \text{sign}(\Sigma), 0), \quad \text{if} \ \ j_x \in [j_x^s, j_x^s + \Sigma), \tag{10}$$

i.e. if the site $(j_x, 0)$ is part of the string. Otherwise

$$g_\Sigma(\boldsymbol{j}) = \boldsymbol{j}, \quad \text{(else)}, \tag{11}$$

except when $\boldsymbol{j} = (j_x^s + \Sigma, 0)$, for which

$$\boldsymbol{g}_\Sigma(j_x^s + \Sigma, 0) = (j_x^s, 0). \tag{12}$$

From this definition, it is easy to see that geometric strings introduce frustrated couplings between spins in squeezed space. The Heisenberg interactions $J\hat{\boldsymbol{S}}_i \cdot \hat{\boldsymbol{S}}_j$ between neighboring sites $\langle i, j\rangle$ in real space can become next-nearest neighbor interactions $J\hat{\boldsymbol{S}}_{\tilde{i}} \cdot \hat{\boldsymbol{S}}_{\tilde{j}}$ in squeezed space, for example, depending on the instantaneous metric $g_\Sigma(\boldsymbol{j})$. Such frustrated couplings introduce additional quantum fluctuations in squeezed space, which lead to local changes of the spin wavefunction $|\psi_0\rangle$ around $\boldsymbol{j}^s$. On the one hand, this can renormalize the string tension $dE/d\ell$. On the other hand, we expect that correlations build up between the spins $\tilde{\boldsymbol{S}}_{\tilde{j}}$ and the string configurations $\Sigma$, in particular when $t$ and $J_{x,y}$ become comparable. Both effects go beyond the frozen spin approximation (FSA) introduced in the main part of the paper. We address them in more detail in a forthcoming work [58].

**Spin-charge correlations with pinned AFM order.** In the main part of the paper, we described how we calculate the two-point function $C_{\text{SH}}^z(d_\text{h})$ defined in Eq. (6) in a three-leg ladder. The result is shown in Fig. 3 (a), for a staggered magnetic field $B = J$ at the short edges, see Fig. 3 (c), pinning the AFM order. Here we present data for different values of the pinning field $B$ and explain how it relates to the theory of geometric strings.

Our results for $0 \leq B \leq J$ are shown in Fig. 7, along with fits to the data at short distances. For the largest pinning field, $B = J$, the behavior of $C_{\text{SH}}^z(d_\text{h})$ is qualitatively similar to the result expected for a hole moving inside a classical Néel state, see Fig. 3 (a). At short distances $d_\text{h}$ of the reference spin in the correlator to the hole, we observe a pronounced dip. As explained in the main text and Fig. 3 (b), this is a direct consequence of the string of displaced spins formed along the trajectory of the hole. When the reference spin at a distance $d_\text{h}$ from the hole approaches the edge of the system, the correlator $C_{\text{SH}}^z(d_\text{h})$ increases for $B = J$ because of the enhanced influence of the pinning field. As shown in Fig. 7, the same behavior is found as long as $B \geq 0.2J$.

For smaller pinning fields $B < 0.2J$, we observe an overall decrease of $C_{\text{SH}}^z(d_\text{h})$ towards smaller values, which is almost independent of $d_\text{h}$. While the shape of the dip at short distances remains almost unaffected, the correlations $C_{\text{SH}}^z(d_\text{h}) < 0$ become negative – first only for small $d_\text{h}$ but eventually everywhere when $B = 0$. This effect is directly related to the sign change of the Néel order observed in Fig. 6, which we interpret as a precursor of stripe formation. Indeed, for sufficiently large $d_\text{h}$ we expect from the FSA introduced in the main text that

$$C_{\text{SH}}^z(d_\text{h}) \approx (-1)^{d_\text{h}} \left[ \langle \hat{S}_{j_0+d_\text{h}}^z\rangle\langle\hat{n}_{j_0}^\text{h}\rangle - \langle\hat{S}_{j_0+1+d_\text{h}}^z\rangle\langle\hat{n}_{j_0+1}^\text{h}\rangle \right] \tag{13}$$

factorizes; I.e. $C_{\text{SH}}^z(d_\text{h}) \simeq \langle\hat{n}_{j_0}^\text{h}\rangle\Omega_{j_0+d_\text{h}}$ reflects the local Néel order parameter $\Omega_{\boldsymbol{j}} = (-1)^{j_x+j_y}\langle\hat{S}_{\boldsymbol{j}}^z\rangle$.

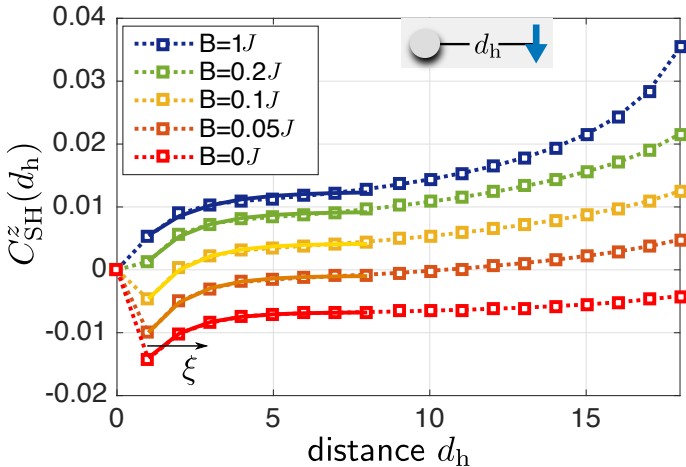

Figure 7: **Pinning long-range order**. We use DMRG to calculate the two-point function $C_{\mathrm{SH}}^z(d_{\mathrm{h}})$, see Eq. (6), in a $40 \times 3$ system with open boundary conditions for $j_0 = L_x/2$. The procedure is identical to the one described in Fig. 3 (a) of the main text, but we tune the strength of the staggered magnetic field $B$ which is applied at the short edges of the system to pin the AFM order, see Fig. 3 (c). The two-point function shows a pronounced dip at short distances $d_{\mathrm{h}}$, which depends only weakly on the applied pinning field $B$. To extract the characteristic length scale associated with the dip, we performed fits of the form $A_1 + A_2 e^{-d_{\mathrm{h}}/\xi(B)}$ in the range $1 \le d_{\mathrm{h}} \le 8$ (solid lines).

As illustrated in Fig. 6 (c), we expect that the holon in the central chain is bound to the two spinon excitations in the upper and lower chains of the three-leg system when $B = 0$. The holon motion still creates geometric strings, connecting the holon to the two spinons in this case. Because the string tension expected from the FSA introduced in the main part of the paper only depends on the local spin correlators of the surrounding spin system, we expect that the length of geometric strings in the case $B = 0$ is similar to the result for a simpler meson with one holon and one spinon in an infinite system.

To confirm our expectation, we extract the characteristic length scale of the dip observed in $C_{\mathrm{SH}}^z(d_{\mathrm{h}})$ at small distances $d_{\mathrm{h}}$. To this end we fit the data with an offset decaying exponential, see Fig. 7. The resulting decay length $\xi(B)$ of these fits is plotted as a function of $B$ in Fig. 3 (d). As concluded in the main part of the paper, this length scale depends only weakly on $B$. Finally, we note that the microscopic origin of the dip observed in $C_{\mathrm{SH}}^z(d_{\mathrm{h}})$ can be different at different values of $B$, but it is always caused by the spinon excitation(s) defining the end of the geometric string.

**Revealing string order**. Here we describe how we identify strings from snapshots of the quantum mechanical wavefunction when measured in the $z$-basis of the spins. Numerically, we generate such snapshots from the ground state wavefunction $|\Psi_{\mathrm{mD}}\rangle$ which we obtain using exact diagonalization of a $6 \times 3$ mixD $t - J$ model with periodic boundary conditions in $x$-direction. To this end we represent $|\Psi_{\mathrm{mD}}\rangle = \sum_\alpha c_\alpha |\alpha\rangle$ in the Fock basis of spins in $z$-direction $\{|\alpha\rangle\}$ and sample Fock states according to the probability distribution defined by $|c_\alpha|^2$.

For a given Fock state $|\alpha\rangle$, we calculate the correlators $C_\sigma(j_x, \alpha) = \langle \alpha | \hat{C}_\sigma(j_x) | \alpha \rangle$ and $C_{\Sigma}^\pm(j_x, \alpha) = \langle \alpha | \hat{C}_{\Sigma}^\pm(j_x) | \alpha \rangle$ defined in Eqs. (7), (8) of the main text, see also Fig. 4 (a). This is easy because the operators $\hat{C}_\sigma$, $\hat{C}_{\Sigma}^\pm$ are diagonal in the $z$-basis. For sites next to the hole, $j_x = j_x^{\mathrm{h}} \pm 1$, we add $-1/4$ to all correlators such that bonds involving the hole contribute as if they were part of a perfect string. This allows us to treat all sites on the same footing in the

following.

In order to determine the string configuration in a snapshot $\alpha$ we apply the following rules, explained in more detail below, to all lattice sites $j_x$ which are not occupied by the hole:

(i) If $C_\sigma(j_x, \alpha) = 0$ and $C_\Sigma^+(j_x, \alpha) = C_\Sigma^-(j_x, \alpha) = +1$, we count $j_x$ as part of a string.

(ii) If $C_\Sigma^+(j_x, \alpha) = C_\Sigma^-(j_x, \alpha) = 0$ and $C_\sigma(j_x, \alpha) = +1$, we count $j_x$ as part of the background (not the string).

(iii) Otherwise, we determine the smallest of the three correlators. If the minimum is realized by $C_\Sigma^-(j_x, \alpha)$ or $C_\Sigma^+(j_x, \alpha)$ or both of them, we count $j_x$ as part of a string. If the minimum is realized by just $C_\sigma(j_x, \alpha)$, we count $j_x$ as part of the background (not the string). In all remaining cases, the configuration remains undefined.

From the so-determined configuration we can extract the string length $\ell(\alpha)$. To this end we start from the site $j_x^{\text{h}}$ occupied by the hole. If the configuration at site $j_x^{\text{h}} + 1$ is a string and the one at site $j_x^{\text{h}} - 1$ is not, or vice-versa, we say that a string of length $\ell \geq 1$ emerges from the hole. We follow it and count the number of segments $\ell(\alpha)$, stopping as soon as one element is no longer found to be in a string configuration. In all other cases, i.e. when no string is present, we set $\ell(\alpha) = 0$. By sampling Fock states $|\alpha\rangle$ as described above, we obtain the string length distributions shown in Fig. 4.

Finally, we explain the motivation for using rules (i) - (iii) defined above. As mentioned in the main text, the basic idea is that nearest-neighbor correlations $C_1$ are enhanced compared to $C_{2,3,\ldots}$ in the 2D Heisenberg AFM due to a large admixture of local singlets [44]. Therefore, if we consider the ground state $|\Psi_0\rangle$ of the 2D Heisenberg model without any geometric strings, $\langle \hat{C}_\sigma(j_x) \rangle = -0.45$ is large and negative while $\langle \hat{C}_\Sigma^\pm(j_x) \rangle = -0.09$ is a five times smaller negative number. In contrast, if we shift all spins on the central chain (with $j_y = 0$) by one lattice site, mimicking the effect of a geometric string, we find that $\langle \hat{C}_\Sigma^\pm(j_x) \rangle$ have large negative values of $-0.45$ and $-0.34$ whereas $\langle \hat{C}_\sigma(j_x) \rangle = -0.09$ is a small negative number much closer to zero. This explains rule (iii), which identifies strings by finding the smallest of the three correlators.

Rules (i) and (ii) are motivated by the effects of quantum fluctuations on top of a classical Néel state pointing in $z$-direction. In a classical Néel state without quantum fluctuations, rule (iii) is sufficient to identify all strings. The two dominant types of quantum fluctuations correspond to flips of individual spins and exchanges of two anti-aligned spins. Rules (i) and (ii) take into account cases where only the central spin at site $(j_x, 0)$ is flipped. As a result $C_\sigma$ changes from $-1$ in the classical Néel state without a string to $+1$, whereas $C_\Sigma^\pm = 0$ remains unchanged. Moreover, $C_\Sigma^\pm$ changes from $-1$ in the classical Néel state with a string at site $(j_x, 0)$ to $+1$. These cases are taken into account by rules (i) and (ii).

**Dimensional crossover.** In the following we describe our analysis of the dimensional crossover from 1D, realized for $J_y = 0$, to the mixD case with $J_y = J_x$. As described in the main text, we performed DMRG simulations in a three-leg ladder with $L_x = 40$ sites. We assumed open boundary conditions, used $t = 3J_x$, set $S^z = 1/2$ and varied $J_y$ between 0 and 1. To extract the string length $\xi(J_y)$ shown in Fig. 5 (a), we calculated the two-point function $C_{\text{SH}}^z(d_{\text{h}})$ defined in Eq. (6) of the main text. The result is shown in Fig. 8, together with the fits by a function $A_1 + A_2 e^{-d_{\text{h}}/\xi}$ which we performed in the range $1 \leq d_{\text{h}} \leq 9$.

In the main text we compare our DMRG results in the three-leg ladder to calculations using the trial wavefunction from Eq. (4) in a $16 \times 8$ system with periodic boundary conditions. To this end we first determined the mean-field spinon Hamiltonian at zero doping and at various $J_y$, for which the Gutzwiller projected mean-field wavefunction $\hat{P}_{\text{GW}}|\Psi_{\text{MF}}\rangle$ has the lowest variational energy, see next paragraph for more details. Then we evaluated the meson wavefunction in Eq. (4) using VMC methods and calculated the three-point function

$C^z_{\mathrm{SHS}}(d, d_{\mathrm{h}} = 1)$. To extract the string length $\xi(J_y)$ shown in Fig. 5, we fitted the result by a function $A_1 + A_2 e^{-d/\xi}$ in the range $3 \leq d \leq 9$. The fits and the data are shown in Fig. 9.

**Mean-field spinon Hamiltonian.** Now we describe the (quadratic) mean-field (MF) Hamiltonian $\hat{\mathcal{H}}_{\mathrm{MF}}$ for the spinons $\hat{f}_{\boldsymbol{k},\sigma}$. We use it to determine the mean-field spinon wavefunction $|\Psi_{\mathrm{MF}}\rangle$ appearing in the trial wavefunction Eq. (4) before Gutzwiller projection. We consider only the half-filling case with zero doping in the following, and treat the couplings defining $\hat{\mathcal{H}}_{\mathrm{MF}}$ as variational parameters which need to be optimized in order to minimize the variational energy $E_0 = \langle\Psi_{\mathrm{MF}}|\hat{\mathcal{P}}_{\mathrm{GW}}\hat{\mathcal{H}}\hat{\mathcal{P}}_{\mathrm{GW}}|\Psi_{\mathrm{MF}}\rangle$. For the isotropic case, $J_x = J_y$, we reproduce exactly the results of Refs. [41, 42].

Following Refs. [40–42] we consider the following class of MF spinon Hamiltonians,

$$\hat{\mathcal{H}}_{\mathrm{MF}} = - t^x_{\mathrm{eff}} \sum_{\langle \boldsymbol{i},\boldsymbol{j}\rangle_x, \sigma} \left( e^{i\theta_{i,j}} \hat{f}^\dagger_{\boldsymbol{j},\sigma} \hat{f}_{\boldsymbol{i},\sigma} + \mathrm{h.c.} \right)$$
$$- t^y_{\mathrm{eff}} \sum_{\langle \boldsymbol{i},\boldsymbol{j}\rangle_y, \sigma} \left( e^{i\theta_{i,j}} \hat{f}^\dagger_{\boldsymbol{j},\sigma} \hat{f}_{\boldsymbol{i},\sigma} + \mathrm{h.c.} \right) + \frac{B_{\mathrm{st}}}{2} \sum_{\boldsymbol{j},\sigma} (-1)^{j_x+j_y} \hat{f}^\dagger_{\boldsymbol{j},\sigma} (-1)^\sigma \hat{f}_{\boldsymbol{j},\sigma}. \quad (14)$$

Here the gauge choice $\theta_{\boldsymbol{i},\boldsymbol{j}} = \Phi/4(-1)^{j_x+j_y+i_x+i_y}$ realizes a staggered magnetic flux $\Phi$ per plaquette, $t^x_{\mathrm{eff}}$ and $t^y_{\mathrm{eff}}$ denote effective NN tunnelings and $B_{\mathrm{st}}$ is an effective staggered magnetic field. The corresponding Bloch Hamiltonian, defined for momenta $\boldsymbol{k}$ in the magnetic Brillouin zone (MBZ), can be written as

$$\hat{\mathcal{H}}_{\mathrm{MF}}(\boldsymbol{k}) = \left( \mathrm{Re}\, r_{\boldsymbol{k}}, \mathrm{Im}\, r_{\boldsymbol{k}}, \frac{(-1)^\sigma B_{\mathrm{st}}}{2} \right)^T \cdot \hat{\boldsymbol{\sigma}}, \quad (15)$$

where $r_{\boldsymbol{k}} = -2t^x_{\mathrm{eff}} \cos(k_x) e^{-i\Phi/4} - 2t^y_{\mathrm{eff}} \cos(k_y) e^{i\Phi/4}$.

In the isotropic case, $J_x = J_y = J$, the optimal parameters were determined in Ref. [42] to be $\Phi = 0.4\pi$ and $B_{\mathrm{st}} = 0.44 \times t^{x,y}_{\mathrm{eff}}$. To study the dimensional crossover where $J_y/J_x$ is varied from 0 to 1, we optimized the mean-field parameters as a function of $J_y$. We observed that the optimum staggered flux $\Phi \approx 0.5\pi$ varies only weakly with $J_y/J_x$, and performed optimization of $B_{\mathrm{st}}/t^x_{\mathrm{eff}}$ and $t^y_{\mathrm{eff}}/t^x_{\mathrm{eff}}$ using a finer grid at fixed $\Phi = 0.5\pi$. The resulting lowest energy $E_0$ is

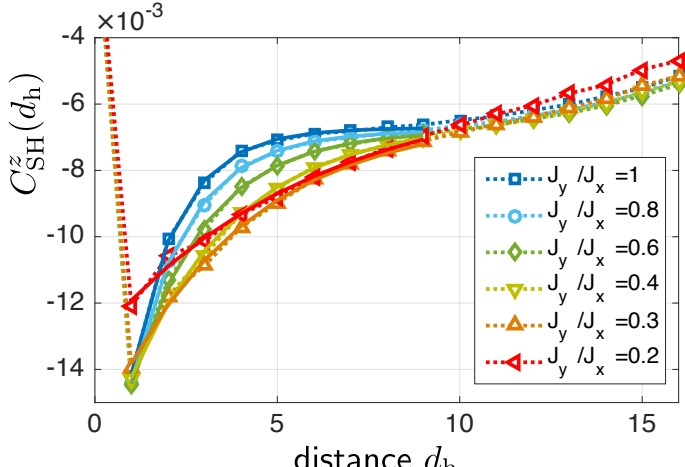

Figure 8: **Spin-hole correlations.** The two-point function $C^z_{\mathrm{SH}}(d_{\mathrm{h}})$ is calculated for different values of $J_y$ using DMRG. Parameters are $L_x = 40$, $L_y = 3$, $S^z = 1/2$ and $t = 3J_x$. The solid lines indicate fits of the form $A_1 + A_2 e^{-d_{\mathrm{h}}/\xi}$ which we used to extract the length scale $\xi(J_y)$.

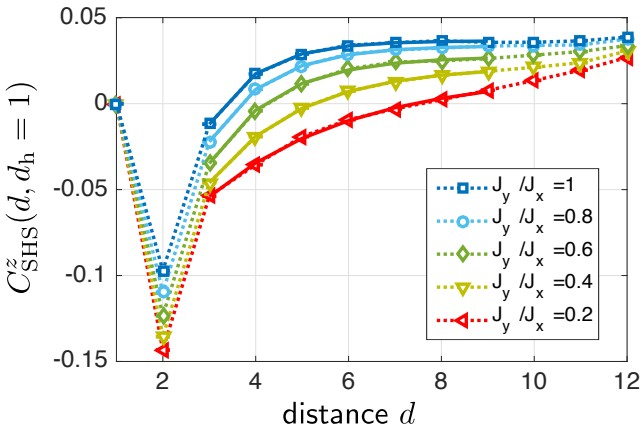

Figure 9: **Spin-hole-spin correlations.** The three-point function $C^z_{\mathrm{SHS}}(d, d_{\mathrm{h}} = 1)$ is calculated for different values of $J_y$ from the meson trial wavefunction in Eq. (4) evaluated at $\boldsymbol{k}_{\mathrm{MP}} = (\pi/2, \pi/2)$, $\Phi = \pi/2$ for optimized $B_{\mathrm{st}}$ as shown in Fig. 10. Parameters are $L_x = 16$, $L_y = 8$, $S^z = 1/2$ and $t = 3J_x$. The solid lines indicate fits of the form $A_1 + A_2 e^{-d/\xi}$ which we used to extract the length scale $\xi(J_y)$.

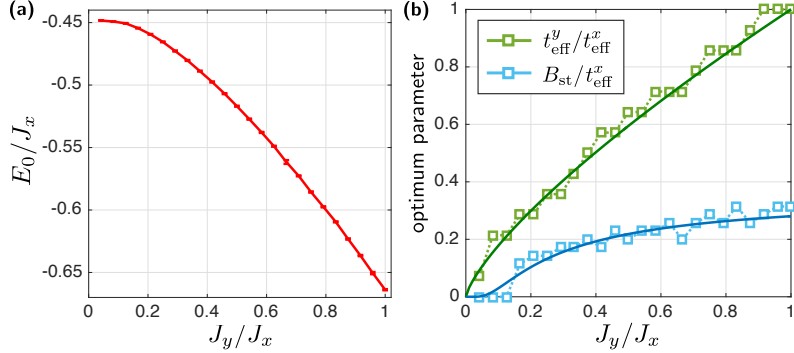

Figure 10: **Dimensional cross over.** (a) The variational energy $E_0$ is shown for a $12 \times 12$ system at zero doping. It was obtained for $\Phi = 0.5\pi$ by minimizing with respect to $t^y_{\mathrm{eff}}/t^x_{\mathrm{eff}}$ and $B_{\mathrm{st}}/t^x_{\mathrm{eff}}$. (b) The optimized parameters at $\Phi = 0.5\pi$ are plotted as a function of $J_y/J_x$. Discrete steps are due to the underlying grid used to determine the optimal variational parameters. The solid lines correspond to Eqs. (16), (17) and provide an approximate description of the optimal values.

shown as a function of $J_y/J_x$ in Fig. 10 (a). The optimal parameters are plotted in Fig. 10 (b). Their numerical values can be approximated by the following curves,

$$B_{\mathrm{st}}|_{\mathrm{opt}} \approx 0.36 \times t^x_{\mathrm{eff}} \, e^{-J_x/(4J_y)}, \tag{16}$$

$$t^y_{\mathrm{eff}}|_{\mathrm{opt}} \approx t^x_{\mathrm{eff}} (J_y/J_x)^{0.75}, \tag{17}$$

which we used in our analysis of the dimensional crossover presented in the main text.

**Data availability.** The data that support the findings of this study are available from the corresponding author upon request.

# Acknowledgements

The authors thank Markus Greiner for suggesting to study the Fermi-Hubbard model in the presence of a strong force along one direction. They are grateful for fruitful discussions with A. Bohrdt, D. Greif, I. Bloch, C. Gross, M. Greiner, T. Hilker, G. Salomon, J. Zeiher, C. Chiu, G. Ji and M. Knap. The authors also acknowledge helpful discussions with S. Todadri, S. Sachdev, Z.-Y. Weng, D. N. Sheng, M. Punk, I. Cirac, L. Vidmar, S. Eggert, P. Zoller, E. Manousakis, M. Kanasz-Nagy, I. Lovas, Y. Wang and R. Schmidt. The authors acknowledge support from Harvard-MIT CUA, NSF Grant No. DMR-1308435, AFOSR Quantum Simulation MURI, the EPiQS program of the Moore foundation. Z.Z. acknowledges his postdoctoral research with Liang Fu at MIT, which is supported by David and Lucile Packard foundation. T.S. acknowledges support by the Thousand-Youth-Talent Program of China.

**Author contributions.** All authors contributed substantially to the writing of the manuscript. F.G. and Z.Z. performed the calculations. F.G., T.S. and E.D. conceived the method.

**Additional information.** The authors declare no competing financial interests.

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
