# Peer review of "Meson formation in mixed-dimensional t-J models"

_SciPost Physics, doi:SciPost Phys. 5, 057 (2018)_

## Round 1 · Referee Report · Anonymous (Referee 1) · 2018-9-28

Strengths

  1. The authors discuss a model which may be useful for exploring the physics of spin-charge separation in 1D in presence of coupling to other degrees of freedom.

  2. A variational wave-function suggested in the paper seems to provide a good qualitative (and even reasonable quantitative description) of spin-charge correlators.

  3. Results could be relevant to cold-atom experiments

Weaknesses

  1. The content of the paper in terms of new physical results seems to be rather confined to the identification of the meson size from the spin-charge correlators, and to some results on dimensional crossover.

  2. The authors address a large number of different points. However, it would have been good to see a more systematic analysis of the results, as well as limitations of their analytical variational approach.

Report

The authors of the paper study the mechanism of spin-charge separation in a one-dimensional t-J model doped with one hole, that is coupled via exchange interactions to a 2D lattice of spins, the latter described by a Heisenberg model. This paper is an extension of some of the authors’ earlier work related to cold-atom experiments measuring hidden antiferromagnetic correlations in a doped 1D Hubbard chain, Ref. [36].

The authors present the result of ED, and DMRG calculations for spin-charge correlation functions, and compare these with the results obtained using variational Monte-Carlo based on a trial wave-function describing a bound state of a spinon and a holon. They find a quantitatively reasonable agreement between these approaches. In addition, the authors study dimensional crossover, and stripe formation using similar approaches, as well as suggest how to identify the bound states in cold atom experiments.

The model suggested in this paper is interesting as it allows one to study how the coupling to spin-background affects the physics of spin-charge separation. Specifically, the authors identify the size of the meson from the spin-charge correlation function calculated using ED and DMRG, and found how meson size scales as the function of inter-channel Heisenberg coupling.

From this perspective, and from its potential relevance to cold-atom experiments, I think this paper is suitable for publication in SciPost, perhaps after some modifications, see questions and comments below.

Requested changes

  1. On page 2 the authors say “… we show that spinons and holons are confined and form bound states”. This statement does not seem to be fully justified or explained.

  2. It would have been useful to see results on the spin-correlators not only along the chain where the hole can hop, but also for the spins in the nearby chains, to see how the hole modifies the spin background.

  3. It would be good to see some tests of the FSA approximation, and specifically a discussion of applicability of the basis states discussed on page 2. Note, that these states are not the ground states of the Heisenberg Hamiltonian with one site removed and will have dynamics.

  4. While the variational wave-function presented in page 2 seems to be reasonable for a 2D system, the results suggest that it also provides a good description even for 3-leg ladders. This is unexpected, and perhaps requires some explanations.

  5. On page 2 the authors cite the values for staggered magnetic flux and staggered field. Do the authors use these same values for all geometries, including finite-size ladders?

  6. For consistency it would have been good to see a comparison with 1D results and for a e.g. a 2-leg ladder, for example in Fig.3.

  7. In Fig.5 it would be good to see the same correlators, and the same system sizes when comparing DMRG and variational MC results. Why does the discrepancy in the ground state energy grows with J_y/J_x?

  8. The paper contains some jargon e.g. “geometric strings”, “squeezed space”, etc. which is not explained clearly. This jargon can also be avoided.

  9. In Fig. 7 the correlators do not seem to converge as a function of the magnetic field, why?

  10. In general, the paper could be shortened, the main statements sharpened and moved to the top of the paper.

  • validity: good
  • significance: ok
  • originality: ok
  • clarity: ok
  • formatting: good
  • grammar: good

Author:  Fabian Grusdt  on 2018-11-19  [id 347]

(in reply to Report 1 on 2018-09-28)

A reply to both referee reports is attached as pdf file. The reply Report 2 starts on page 1 of 12.

Attachment:

RefereeReplyMixD-SciPost_3JVxU45.pdf

---

## Round 1 · Referee Report · Anonymous (Referee 2) · 2018-11-5

Strengths

The authors introduce a simplified model to describe a doped 2D Heisenberg model, which may shed some new light on the persisting open problem and enable researchers to study spin correlations in experimental context.

Weaknesses

The authors should clarify the applicability of the model for higher doping (beyond a single hole) and what are the possible efficient extensions of it.

Report

The authors study the structure of an object (a "meson") when a hole is created in a two-dimensional Heisenberg antiferromagnet. The topic of the doped 2D Heisenberg model is challenging and despite of its origin in 1980's it's still an important open question that needs to be solved. The authors introduce a simplified model, which may shed some new light on the problem and enable researchers to study spin correlations in experimental context. I recommend the publication of the manuscript in SciPost, after the authors address the comments given below.

As far as I understand, the setup and model under investigation is useful to study the "meson" structure. I wonder what else one can learn from the model when going to higher doping (anything beyond a single hole). In particular, in case of more holes there are many possible configurations how to design the "filling" of chains. The subtlety of the competition between superconductivity and the stripe phase, as the authors mention in discussion, is that the emergence of "hole-rich" and "hole-depleted" regions eventually takes place in a homogeneous 2D lattice. In this model, however, the distribution of holes should be done by hand. It hence seems that the model should be modified, e.g., by allowing a perpendicular hopping, but then the numerical difficulty again starts to be a problem. I think the authors should clarify for what exactly, beyond the studies of the "meson" structure, the particular model should be useful, or what are the possible efficient extensions of it.

Requested changes

Looking at the SciPost webpage, the Referee report on 2018/Sept/28 poses some relevant questions that should be addressed in a modified version of the manuscript. Additional comment is related to a better explanation of numerical results in Fig. 2: In (a), why are results from a 6 x 3 cluster plotted as 6 x 6 density plot, in (b), why are results from a 16 x 8 cluster plotted as 9 x 9 plot.

  • validity: good
  • significance: good
  • originality: good
  • clarity: good
  • formatting: excellent
  • grammar: excellent

Author:  Fabian Grusdt  on 2018-11-19  [id 346]

(in reply to Report 2 on 2018-11-05)

A reply to both referee reports is attached as pdf file. The reply Report 2 starts on page 9 of 12.

Attachment:

RefereeReplyMixD-SciPost.pdf

---

## Round 2 · Author Response

Dear Editors,

we would like to resubmit our paper “Meson formation in mixed-dimensional t-J models” to SciPost. We are grateful for the referees careful assessments, and after taking into account their recommendations and criticism, we believe that our work is now ready for publication. A detailed point-by-point response to the referee reports is provided in the pdf file uploaded in response to the referees.

Yours sincerely,
Fabian Grusdt
Zheng Zhu
Tao Shi
Eugene Demler

---

## Round 2 · List of Changes

We have now added a reference to the recent paper [Chiu et al., arXiv:1810.03584] in the discussion at the end of our manuscript.
To address a comment by the referee we have added a sentence in our manuscript on page 2: “We will confirm below, in Fig. 4, that the FSA is a reliable approximation.”
In Fig. 2 b) we have now indicated explicitly in the figure caption the specific parameters.
In Fig. 5 we have included an explicit reference to the Methods section now. We have also added an explanation in the caption of Fig. 9 in the Methods.
On page 2 we have now clarified what we mean by geometric strings: “In the approximate set of basis states constructed so far this corresponds to a displacement of all spins along $\Sigma$, referred to as the geometric string, connecting $j_x$ and $j_x^s$.”
We have included a sentence in discussion section now: “Our model Hamiltonian can be implemented at arbitrary doping levels using ultracold atoms in optical lattices.”
To make the reader aware of the richness of our system at finite doping, we have included a remark on page 7 of our manuscript: “Note that the mixD Hamiltonian has many independent sectors of individually tunable doping levels per chain, which need to be studied separately.”

---

## Editorial Decision

published